# Factors Associated with *Toxoplasma gondii* Seroprevalence in Pregnant Women: A Cross-Sectional Study in Belgrade, Serbia

**DOI:** 10.3390/pathogens12101240

**Published:** 2023-10-13

**Authors:** Ljiljana Marković-Denić, Milena Stopić, Branko Bobić, Vladimir Nikolić, Iva Djilas, Snežana Jovanović Srzentić, Tijana Štajner

**Affiliations:** 1Faculty of Medicine, Institute of Epidemiology, University of Belgrade, 11000 Belgrade, Serbia; nikolicvladimir32@gmail.com; 2National Reference Laboratory for Toxoplasmosis, Group for Microbiology and Parasitology, Center of Excellence for Food- and Vector-Borne Zoonoses, Institute for Medical Research, National Institute of Republic of Serbia, University of Belgrade, 11129 Belgrade, Serbia; milenastopic@yahoo.com (M.S.); bobicb@imi.bg.ac.rs (B.B.); tijana.stajner@imi.bg.ac.rs (T.Š.); 3Blood Transfusion Institute of Serbia, 11000 Belgrade, Serbia; ivaiva@gmail.com (I.D.); srzentics60@yahoo.com (S.J.S.)

**Keywords:** *Toxoplasma gondii*, toxoplasmosis, seroprevalence, risk factors, pregnant women

## Abstract

Toxoplasmosis, caused by the cosmopolitan protozoan *Toxoplasma gondii*, has particular implications during pregnancy due to the possible transmission of infection to the fetus. Very few studies have assessed seroprevalence and the risk factors for toxoplasmosis in healthy pregnant women. The aim of this study was to examine the seroprevalence of *T. gondii* infection in healthy pregnant women and to identify the associated risk factors for toxoplasmosis. The cross-sectional study involved 300 healthy pregnant women who came to the Institute for Blood Transfusion in Belgrade between November 2018 and February 2019 for routine blood group and Rh factor testing before delivery, who were also tested using serological screening for the presence of specific antibodies. Positives were further examined using enzyme immunoassay. Of the total sera of participants analyzed, 38 were positive for specific IgG, resulting in a seroprevalence rate of 12.7% (95% Confidence Interval (CI) 9.1–17.0%). All pregnant women presented negative anti-*T. gondii* IgM antibodies. The multivariate logistic regression analysis revealed that living in a house with a garden was independently associated with the risk of *T. gondii* infections, while eating chicken meat was connected with a lower risk compared to eating other types of meat with an odds ratio (OR) of 2.5 (95% CI 1.21–5.02) and an OR of 0.3 (95% CI 0.09–0.83), respectively. Although the prevalence of anti-*T. gondii* IgG antibodies is relatively low, it is essential to maintain and adapt evidence-based preventive measures for toxoplasmosis continually.

## 1. Introduction

*Toxoplasma gondii*, the cosmopolitan protozoan responsible for toxoplasmosis, stands as one of the most widespread zoonotic infections globally. Due to the high percentage of asymptomatic cases, accurate estimation of the true number of individuals affected by toxoplasmosis is challenging and is based on laboratory methods. Hence, the detection of specific antibodies is the pivotal element of every epidemiological study aimed at determining the seroprevalence of *T. gondii.* A recent comprehensive literature review revealed that the overall seroprevalence ranged from 0.5 to 87.7%, with 25.7% as the average global seroprevalence rate [1]. The variation of *T. gondii* seroprevalence depends on several factors, such as geographic location, environmental conditions, dietary habits, and socio-demographic characteristics [2]. Climatic conditions significantly impact the survival of *T. gondii* parasites in the environment [3]. Toxoplasmosis is most prevalent in tropical countries, such as Brazil and Colombia, where high temperatures and increased humidity create optimal conditions for oocyst maturation and survival outdoors.

In addition, socioeconomic factors, like a country’s economic development stage, have an impact on both toxoplasmosis prevalence and the incidence of primary (acute) toxoplasmosis in pregnant women, as well as congenital toxoplasmosis. Studies conducted worldwide indicate that the global *T. gondii* seroprevalence (specific IgG) among pregnant women was estimated to be 32.9% (95% confidence interval: 7.8–15.1%), with significant regional variation. The highest seroprevalence was found in the region of the Americas (45.2%), followed by the Eastern Mediterranean region (39.7%), Africa (36.5%), Europe (30.0%), the South-East Asia region (24.6%), and the Western Pacific region (11.2%). Each region also had considerable heterogeneity among countries [4]. Notably, there has been a consistent downward trend in toxoplasmosis prevalence across Europe over the past few decades, including the Balkan countries, and among them Serbia [5,6,7].

Notably, Serbia has experienced a dramatic decline in toxoplasmosis prevalence, dropping from 86% in 1988 to 32.5% in 2007 [8,9]. Initially, the prevalence in Serbia was 50% from 1960 to 1980 but began rising in the 1980s, possibly due to the introduction of new serological tests, before steadily declining since the 1990s, mirroring the trend observed in Europe [9]. The first study conducted in Belgrade from 1988 to 1991 revealed that the leading risk factor for toxoplasmosis was the consumption of undercooked meat [10]. The most recent research on women of childbearing age conducted from 2001 to 2005 indicated that undercooked meat was still the primary risk factor for acutely infected women [9]. Cats, as definitive hosts, are a source of infection through environmental contamination [11]. Direct contact with cats is not a common route of transmission, unlike contact with cat feces. Contaminated soil is a significant risk factor for *T. gondii* infection. A European multicenter study indicated that 6 to 17% of toxoplasmosis cases resulted from human contact with contaminated soil. In aquatic environments, oocysts can survive for extended periods [12].

In Serbia, preventive measures are oriented toward educating women of childbearing age about the known risk factors. In such preventive programs, the emphasis is on the consumption of adequately cooked meat, thorough washing of fruits and vegetables before consumption, handwashing after soil contact, and the use of appropriate protective gear when working in gardens. Pregnant women are advised to avoid cleaning cat litter boxes during pregnancy or to use gloves when doing so.

While acute infection in pregnant women is usually asymptomatic (over 80%), unless timely diagnosed and treated, transplacental transmission of the parasites may occur anytime, at rates mainly dependent on the stage of the pregnancy and virulence of the *T. gondii* strain.

Maternal infection during the first trimester rarely results in fetal infection due to a low rate of transplacental transmission in early pregnancy. However, when it does occur, it can lead to severe clinical manifestations in the fetus (such as hydrocephalus, chorioretinitis, and neurological impairment) and even result in miscarriage or stillbirth. In the second and third trimesters, the transmission rates steadily increase (25–50% and 50–70%, respectively). Even though the majority of infected newborns are asymptomatic at birth, severe sequelae, in particular chorioretinitis, leading to visual impairment, manifest at any time later in life [13,14,15]. Very few studies of seroprevalence and the risk factors for toxoplasmosis determined in a population of healthy pregnant women without any signs of infection deprive their results of any selection-related bias, which is often present in epidemiological studies focusing on pregnant women. Therefore, we conducted a comprehensive epidemiological study among healthy pregnant women, with no signs or symptoms of infection, to ascertain the seroprevalence of *T. gondii* infection and to identify associated risk factors for toxoplasmosis.

## 2. Materials and Methods

### 2.1. Study Population and Data Collection

The cross-sectional study involved 300 healthy pregnant women from Belgrade. The sample consisted of consecutive women attending the Institute for Blood Transfusion of Serbia for routine blood group and Rh factor testing before delivery during the period between November 2018 and February 2019. Women were eligible to participate if they were over 15 years old. The written informed consent form prepared by researchers and approved by the Ethics Committee was filled out by every pregnant woman who agreed to participate in the research. Blood sampling and an epidemiological interview of the pregnant women was conducted at the Institute for Blood Transfusion of Serbia. The analytical part of the study (research planning and laboratory tests) was performed at the National Reference Laboratory for Toxoplasmosis, the Institute for Medical Research, University of Belgrade.

After informed consent, the pregnant women were interviewed by a physician using an epidemiological questionnaire specifically designed for this type of study to detect potential risk factors for *T. gondii* infection. The questionnaire was divided into a four sections: (1) the basic demographic data: gender, age, level of education, residence, data on type of housing; (2) data on lifestyle habits: consumption of meat, consumption of raw/undercooked meat, type of meat (chicken, pork, lamb, or beef), contact with a cat, and contact with soil; (3) data on previous results of *T. gondii* infection tests if they were performed for medical or personal reasons; (4) data from gynecological-obstetrical anamnesis on previous pregnancy outcomes (spontaneous abortion, premature birth, and stillbirth). Apart from basic demographic data and data on the type of meat consumed, all the questions were dichotomous (yes or no). Accordingly, we considered the respondents to be exposed to a risk factor if they gave an affirmative answer.

### 2.2. Determining the Status of T. gondii Infection

A high-sensitivity direct agglutination assay (HSDA) was used for the serological screening of the presence of specific IgG antibodies. Serological screening for the presence of specific IgG antibodies was performed with an in-house high sensitivity direct agglutination assay (HSDA) using formalin-fixed tachyzoites of the *T. gondii* RH strain as the antigen as previously described [16]. All the sera were serially diluted twofold starting from 1:20. Visible agglutination at a serum dilution of 1:40 was considered a positive result. Each sample in which specific IgG antibodies were detected using HSDA was further analyzed with a commercial enzyme immunoassay (VIDAS^®^ Toxo IgG II, TXG, *Biomerieux*, Marcy-l’Étoile, France). The concentration of specific IgG was interpreted according to the manufacturer’s instructions (<4 IU/mL negative, 4–8 IU/mL borderline and ≥8 IU/mL positive). All samples with specific IgG ≥ 8 IU/mL were further examined for their avidity, using a VIDAS^®^ Toxo IgG avidity assay (TXGA, *Biomerieux*, France). The result was expressed as an avidity index (IA) and interpreted according to the manufacturer’s instructions, noting that high IA excludes *T. gondii* infection in the last 4 months (IA < 0.2 low avidity, IA 0.2–0.3 borderline and IA > 0.3 high avidity). If the TXGA showed low or borderline avidity, specific IgM antibodies were determined using enzyme immunoassay (VIDAS^®^ Toxo IgM, TXM, *Biomerieux*, France), in order to confirm or exclude acute infection. The result was expressed in the form of an index (I) and interpreted according to the manufacturer’s instructions (I < 0.55 negative, 0.55–0.65 borderline and >0.65 positive).

Samples that were positive in the HSDA were further examined using enzyme immunoassay for the determination of specific IgG antibodies in an automated mini VIDAS (VIDAS^®^ Toxo IgG II, TXG, *Biomerieux,* France). The status of *T. gondii* infection was assessed if the concentration of specific IgG antibodies in the sera was ≥8 IU/mL by determining the avidity of specific IgG antibodies using an enzyme immunoassay (VIDAS^®^ Toxo IgG avidity TXGA, *Biomerieux,* France). In pregnant women with a low (<0.200) or borderline (0.200–0.300) index avidity of specific IgG antibodies, specific IgM antibodies were also determined using enzyme immunoassay (VIDAS^®^ Toxo IgM, TXM, *Biomerieux,* France) to confirm/exclude primary infection. A high avidity index (>0.300) excluded infection in the last four months.

### 2.3. Statistical Analysis

The data were statistically analyzed using adequate methods of descriptive and analytical statistics. The data were expressed as means ± SD for continuous variables, and numbers (percentages) were used for categorical variables. The *T. gondii* infection risk factors were analyzed using logistic regression. Variables that proved to be significant in univariate regression models at a significance level of *p* < 0.1 were included in the multivariate logistic regression model, with *T. gondii* infection as the outcome variable. The results of the multivariate regression model were presented as an odds ratio (OR) with a 95% confidence interval (95% CI) and the Hosmer–Lemeshow goodness-of-fit test was performed to assess the overall model fit. The statistical analyses were performed in the statistical package SPSS for Windows 23.0 (SPSS Inc., Chicago, IL, USA).

## 3. Results

Generally considered a leading marker of acute infection, *T. gondii*-specific IgM antibodies were absent in all pregnant women analyzed during our study. In total, 38/300 women presented with *T. gondii* specific IgG, resulting in a seroprevalence rate of 12.7% (95% CI 9.1–17.0%).

Table 1 presents the demographic data for the pregnant women alongside the univariate logistic regression results for *T. gondii* infection. The age range of the participants was from 20 to 44 years old, with a median of 31 years. Although the majority (81.6%) of the 38 pregnant women with specific IgG antibodies against *T. gondii* were classified within the 25- to 34-years-old category, no significant association was observed between age and *T. gondii* infection. Pregnant women living in a house with a garden were more likely to be infected than those residing in flats, with an odds ratio of 2.3.

Table 2 contains data on the lifestyle habits potentially associated with contracting *T. gondii* infection. Interestingly, neither contact with soil nor contact with a cat were associated with T. gondii infection in the examined pregnant women. Furthermore, pregnant women who consumed chicken meat were less likely to be infected than those who consumed beef (OR = 0.3).

Among *T. gondii* seropositive pregnant women, 63.6% were in their first pregnancy, and half (50%) of the seropositives were in their third trimester, although these factors did not reach statistical significance. Furthermore, 94.3% of pregnant women were currently experiencing normal pregnancies, while a history of spontaneous abortion was reported in 15.3% of pregnant women, with no statistically significant difference observed (*p* = 0.236 and *p* = 0.168, respectively). Additionally, a history of preterm birth or stillbirth was not identified as a risk factor for *T. gondii* seropositivity. The gynecological and obstetric history of pregnant women is presented in Table 3.

Table 4 presents data related to multivariate logistic regression, revealing that living in a house with a garden was an independent risk factor while eating chicken meat was identified as an independent protective factor against *T. gondii* infection with odds ratios of 2.5 and 0.3, respectively.

## 4. Discussion

The prevalence of *T. gondii* among pregnant women in our study was 12.7%, which is considerably lower than ever before noted in Serbia, in both pregnant/childbearing age women and the general population.

According to the results of a recently published systematic review and meta-analysis [4], the highest estimated *T. gondii* (specific IgG) seroprevalence based on empirical data among pregnant women was observed in Ethiopia (64%), Gabon (57%), and Brazil (54%). The lowest seroprevalence was in Mexico (7%), South Korea (2%), and Canada (0.2%). Among WHO regions, the Americas were designated the highest prevalence (45%), whereas the Western Pacific was designated the lowest (11%). With an average IgG seroprevalence of 30%, Europe belongs to vulnerable regions, especially in terms of infection in pregnancy and congenital toxoplasmosis [4]. However, there are significant differences between regions and countries due to variations in environmental conditions and dietary and lifestyle habits. In general, *T. gondii* prevalence is lower in high-income countries. Indeed, seroprevalence among pregnant women is generally low in Northern and Western European countries: 9% in Norway [17], 18% in Sweden [18], and 18% in the Netherlands [19]. In France, a country with a decades-long screening program for toxoplasmosis in pregnancy, seroprevalence among pregnant women was 54% in 1995. Although a continuously decreasing trend was observed after this period, this prevalence is still relatively high, 31% in 2016, according to data from national perinatal surveys [20]. Traditionally, higher seroprevalence was noted in the Southern and Eastern parts of Europe. However, the overall prevalence in pregnant women recently recorded in Italy was 22%; 18% in native Italians and 33% in immigrants [21]. The *T. gondii* seroprevalence obtained in our study was indeed similar to the more developed European countries, unlike the infection rates detected in our neighboring countries. For instance, in Romania, recent seroprevalence data in pregnant women of 38.8%, although slightly lower than the 43.8% reported 10 years ago [5], were still significantly higher than in Serbia nowadays. Furthermore, the seropositivity rate in pregnant women in Bulgaria during the 2011–2020 period was 19% [22], lower than in Romania though still much higher than our 12.7%.

A crucial measure of secondary prevention, serological testing for *T. gondii* infection status in pregnant women, involves detecting the specific IgM and IgG antibodies at the beginning of pregnancy and during each trimester. If seroconversion occurs during pregnancy, prenatal CT diagnosis is recommended. An example of an effective serological screening program for pregnant women is in France, with monthly toxoplasmosis testing until delivery. In this country, the prevalence of congenital toxoplasmosis was highest from 1987 to 1991. Still, it significantly decreased from 1992 to 2008, confirming the justification for a screening program in countries with high *T. gondi* seroprevalence. In addition to France, five other EU countries (Austria, Belgium, Greece, Slovakia, and Slovenia) have compulsory screening of pregnant women. In contrast, in four countries (Bulgaria, Czechia, Germany, and Hungary), screening is voluntary [23]. Austria and Slovenia conduct serological screening of pregnant women once per trimester. The introduction and design of a serological screening program depend on its presumed cost-benefit impact. Italy has recently introduced screening, while in some countries (Belgium, Switzerland, and Lithuania), screening programs have been in place for several years. However, in certain European countries (the United Kingdom and Norway), the costs of potential screening programs are not considered justified given the existing seroprevalence of toxoplasmosis among pregnant women [24]. Serological screening for *T. gondii* in pregnancy is not mandatory in our country and is not routinely offered to pregnant women as part of their antenatal care. However, ToRCH testing is available at our own expense in our country.

According to our results, the factors independently associated with *T. gondii* among pregnant women include living in a house with a garden, while eating chicken meat was connected with lower risk compared to eating other types of meat. Living in a house with a garden was undoubtedly associated with more frequent contact with soil during work in a garden or orchard. The pregnant women included in our study more often lived in such housing conditions. Although the data on contact with the soil as an independent risk factor did not reflect a significant difference in seropositivity, living in a house with a garden is an indirect indicator of contact with the soil and probably with cats. As cats are the only source of oocysts that can contaminate the soil, our results suggest the possibility that a significant number of cats in this region are infected with *T. gondii*. Street cats significantly impact soil contamination, but the impact of domestic cats should not be neglected either. It is uncertain whether every instance of contact with contaminated soil will cause infection. It depends primarily on the number of oocysts that have reached the ground through cat feces, and it is determined by the incidence of *T. gondii* infection in cats and the density of the area’s population of these animals [25]. Excreted oocysts are highly resistant, and once sporulated, can survive for up to two years in a suitable environment [26]. The transmission of oocysts via accidental ingestion of soil can occur during gardening or the consumption of homegrown vegetables. There are efforts to design a quantitative risk assessment model for estimating the risk of *T. gondii* infection via soil [27].

Our results showed that chicken consumption was associated with a lower rate of *T. gondii* infection compared to lamb, pork, or more than one type of meat. The consumption of chicken meat can be a source of infection, primarily if chickens are raised on pastures using the so-called “free-range method” because they can come into contact with oocysts more easily than when they are grown on farms in closed, controlled systems [28]. In the last few decades, the consumption of meat originating from free-range farming has increased in Europe. Therefore, the prevalence of *T. gondii* infection in animals reared in free-range systems compared to those raised in conventional (closed) systems could in time be reflected in the prevalence of *T. gondii* infection in humans [29]. In Serbia, there are no recent data on *T. gondii* in chickens or chicken meat because the previous research efforts were mainly, as was the case worldwide, based on other types of meat (beef, pork, and lamb) [9]. There are, however, scarce data from a study conducted 20 years ago [30]. In general, studies have reported a low concentration of *T. gondii* parasites in chicken organs. In poultry, the antibody levels decrease one month after infection. This can lead to underestimation of the true prevalence of *T. gondii* in chickens and turkeys [31]. Therefore, proper cooking of poultry meat still stands out as the crucial preventive measure [32].

The majority of pregnant women we found to be seropositive with *T. gondii* (81.6%) belonged to the 25–34 age group. However, we did not find a significant association between age and *T. gondii* infection. Our results are in concordance with the findings from Italy [21,33] and Romania [34]. In some countries with very high *T. gondii* seroprevalence, a more significant percentage of seropositive pregnant women are of advanced age [35,36]. In contrast, in others, no significant association was observed between seropositivity and age [37]. In Europe, a positive correlation between seroprevalence and age was found in France. In this country, seroprevalence in pregnant women increased from 19.5% among women 20–24 years to 51.7% among women older than 40 years [20]. A similar linear increase with age was found in Poland [38]. This association is probably due to more prolonged exposure to the *T. gondii* infection risk factors, especially in countries with high seroprevalence.

An association between *T. gondii* infection and the outcome of previous pregnancies (spontaneous abortion, premature birth, and stillbirth) was not found in our study, similar to other studies in which there was no connection between toxoplasmosis and pregnancy outcome [39,40].

This research has several limitations. Besides the well-known limitations of the design of a cross-sectional study, the choice of study place was a relevant limitation. Only the population of pregnant women in Belgrade, the capital, was included. Access to healthcare and the quality of health services are usually reduced outside the capital, especially in suburban areas of the country; hence, the seroprevalence rate could be higher in those areas. Irrespective of the advantages of multicenter studies, we decided to conduct the study in one center, the capital, due to our limited resources. The Belgrade municipality encompasses both urban and rural areas, resulting in a population with varying levels of education and habits that could represent pregnant women from other parts of our country. Cities and suburban municipalities are well connected by road, and people’s transportation is quite simple. Therefore, we avoided the healthcare access bias that occurs when subjects have decreased access to healthcare. Furthermore, the precise timing of infection could not be established using the available laboratory-based methods in chronically infected pregnant women. Therefore, the exposure to various risk factors and specific IgG positivity rates do not necessarily indicate a causal relationship. Also, the risk factor assessment in our study was based on the participants’ responses. Recall bias is often a shortcoming of many studies. However, this is usually not the case with studies involving pregnant women, as their awareness of the potential risk factors, with the responsibility of being the sole caregivers for their fetus/newborn, is usually high.

## 5. Conclusions

Living in a house with a garden is a risk factor for *T. gondii* infection. Pregnant women who consumed chicken meat were less likely to be infected than those who consumed beef. Although the *T. gondii* seroprevalence is relatively low in our country, there is a need for the continuous use of proven preventive measures against toxoplasmosis. The focus should be on pregnant women’s education with specific exposition on environmental risk factors as the primary prevention method and prenatal screening as secondary prevention.

## Figures and Tables

**Table 1 pathogens-12-01240-t001:** Demographic characteristics of pregnant women and univariate logistic regression analysis for *T. gondii* infection.

	Total n (%)	*T. gondii* Seronegative n (%)	*T. gondii* Seropositive n (%)	Univariate Logistic Regression OR (95% CI)	*p* Value
**Age group** 20–24 years 25–29 years 30–34 years 35–40 years >40 years	300 (100) 27 (9.0) 78 (26.0) 132 (44.0) 44 (14.7) 19 (6.3)	262 (100) 23 (8.8) 65 (24.8) 114 (43.5) 42 (16.0) 18 (6.9)	38 (100) 4 (10.5) 13 (34.2) 18 (47.4) 2 (5.3) 1 (2.6)	Ref. 1.2 (0.34–3.89) 0.9 (0.28–2.93) 0.3 (0.05–1.61) 0.3 (0.03–3.11)	0.822 0.872 0.152 0.326
**Level of education** Elementary school High school University	280 (100) 18 (6.4) 117 (41.8) 145 (51.8)	245 (100) 16 (6.5) 104 (42.4) 125 (51.0)	35 (100) 2 (5.7) 13 (37.1) 20 (57.1)	Ref. 1.0 (0.20–4.85) 1.3 (0.27–5.99)	1.00 0.754
**Residence area** Suburban Urban	300 (100) 31 (10.3) 269 (89.7)	262 (100) 24 (9.2) 238 (90.8)	38 (100) 7 (18.4) 31 (81.6)	2.2 (0.89–5.62)	0.086
**Type of housing** Townhouse with garden Flat in building	300 (100) 113 (37.7) 187 (62.3)	262 (100) 92 (35.1) 170 (64.9)	38 (100) 21 (55.3) 17 (44.7)	2.3 (1.15–4.54)	**0.019**

**Table 2 pathogens-12-01240-t002:** Lifestyle habits of pregnant women and univariate logistic regression analysis for *T. gondii* infection.

Lifestyle Habits	Total n (%)	*T. gondii* Seronegative n (%)	*T. gondii* Seropositive n (%)	Univariate Logistic Regression OR (95% CI)	*p* Value
**Contact with soil** Yes No	300 (100.0) 74 (24.7) 226 (75.3)	262 (100.0) 60 (22.9) 202 (77.1)	38 (100.0) 14 (36.8) 24 (63.2)	1.9 (0.96–4.03)	0.066
**Contact with cat** Yes No	300 (100.0) 69 (23.0) 231 (77.0)	262 (100.0) 59 (22.5) 203 (77.5)	38 (100.0) 10 (26.3) 28 (73.7)	1.2 (0.56–2.68)	0.604
**Owner of cat** Yes No	300 (100.0) 22 (7.3) 278 (92.7)	262 (100.0) 17 (6.5) 245 (93.5)	38 (100.0) 5 (13.2) 33 (86.7)	2.2 (0.76–6.13)	0.149
**Contact with outdoor cat** Yes No	300 (100.0) 55 (18.3) 245 (81.7)	262 (100.0) 48 (18.3) 214 (81.7)	38 (100.0) 7 (18.4) 31 (81.6)	1.0 (0.42–2.42)	0.988
**Consumption of meat** Yes No	300 (100.0) 297 (99.0) 3 (1.0)	262 (100.0) 259 (98.9) 3 (1.1)	38 (100.0) 38 (100.0) 0 (0)	0.4 (0.04–4.23)	0.468
**Consumption of raw/undercooked meat** Yes No	291 (100.0) 82 (28.2) 209 (71.8)	253 (100.0) 73 (28.9) 180 (71.1)	38 (100.0) 9 (23.7) 29 (76.3)	0.8 (0.35–1.70)	0.510
**Type of meat** Beef Pork Lamb Chicken More than one type	297 (100.0) 22 (7.4) 59 (19.9) 2 (0.7) 175 (58.9) 39 (13.1)	259 (100.0) 16 (6.2) 51 (19.7) 1 (0.4) 158 (61.0) 33 (12.7)	38 (100.0) 6 (15.8) 8 (21.1) 1 (2.6) 17 (44.7) 6 (15.8)	Ref. 0.4 (0.13–1.39) 2.7 (0.14–49.76) 0.3 (0.10–0.83) 0.5 (0.13–1.74)	0.154 0.511 **0.021** 0.267

**Table 3 pathogens-12-01240-t003:** Gynecological and obstetric history of pregnant women and univariate logistic regression analysis for *T. gondii* infection.

	Total n (%)	*T. gondii* Seronegative n (%)	*T. gondii* Seropositive n (%)	Univariate Logistic Regression OR (95% CI)	*p* Value
**First pregnancy**	282 (100.0)	249 (100.0)	33 (100.0)		
Yes	151 (53.5)	130 (52.2)	21 (63.6)	1.6 (0.76–3.39)	0.219
No	131 (46.5)	119 (47.8)	12 (36.4)		
**Pregnancy trimester**	296 (100.0)	258 (100.0)	38 (100.0)		
I	72 (24.3)	59 (22.9)	13 (34.2)	Ref.	
II	57 (19.3)	51 (19.8)	6 (15.8)	0.5 (0.19–1.51)	0.236
III	167 (56.4)	148 (57.4)	19 (50.0)	0.6 (0.27–1.25)	0.168
**Normal pregnancy**	194 (100.0)	169 (100.0)	25 (100.0)		
Yes	183 (94.3)	160 (94.7)	23 (92.0)	0.5 (0.19–1.51)	0.236
No	11 (5.7)	9 (5.3)	2 (8.0)		
**Spontaneous abortion history**	274 (100.0)	242 (100.0)	32 (100.0)		
Yes	42 (15.3)	39 (16.1)	3 (9.4)	0.6 (0.27–1.25)	0.168
No	232 (84.7)	203 (83.9)	29 (90.6)		
**Preterm birth history**	275 (100.0)	243 (100.0)	32(100.0)		
Yes	8 (2.9)	8 (3.3)	0 (0.0)	NA	0.999
No	267 (97.1)	235 (96.7)	32 (100)		
**Stillbirth history**	273 (100.0)	241 (100.0)	32(100.0)		
Yes	4 (1.5)	3 (1.2)	1 (3.1)	2.6 (0.26–25.37)	0.422
No	269 (98.5)	238 (98.8)	31 (96.7)		

**Table 4 pathogens-12-01240-t004:** Multivariate logistic regression of risk factors for *T. gondii* infection among pregnant women.

Variable	Multivariate Logistic Regression OR (95% CI)	*p* Value
**Type of housing**		
Townhouse with garden	2.5 (1.21–5.02)	**0.013**
Flat in building		
**Type of meat**		
Beef	Ref.	
Pork	0.4 (0.11–1.23)	0.104
Lamb	3.9 (0.20–73.66)	0.370
Chicken	0.3 (0.94–0.83)	**0.021**
More than one type	0.5 (0.13–1.79)	0.279

## Data Availability

The data presented in this study are available on request from the corresponding author. The data are not publicly available due to [privacy].

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
