# Peer review of "Factors Associated with Toxoplasma gondii Seroprevalence in Pregnant Women: A Cross-Sectional Study in Belgrade, Serbia"

_pathogens, 2023, doi:10.3390/pathogens12101240_

Round 1
Reviewer 1 Report
General comments
The study is well-written and provides valuable information on the seroprevalence and associated risk factors of T. gondii antibodies in pregnant women in Belgrade, Serbia. However, there are several points need clarification from the authors such as;
- In the title, the authors should specify Belgrade in the title, because it is the only tested region in Serbia.
- In abstract and introduction, the authors should clearly highlight the point of novelty of this study.
-In introduction, the authors should prepare a paragraph for previous reports on seroprevalence and risk factors analysis of T. gondii either in human or animals.
- In methods, the authors need to add some details on type of sample used and methods of preparation.
- In limitations at lines 214-255, why do authors investigate only pregnant women in Belgrade and what are the efforts done to make this study more comprehensive?
Minor points
Some typos errors corrections are required as in the following points;
- In title (line 2), the word “Toxoplasma gondii” should be changed to italic, and this should be checked throughout the manuscript.
- Line 23, this “95%CI95 9.1-17.0%” should be corrected and full name should be written because it is the first appearance..
- Line 27, OR should be written in full name because it is the first appearance.
- Line 51, regarding this expression “95% credible interval” did the author mean confidence interval?
No serious issues regarding the English edition. Minor revision is required.
Author Response
Dear reviewer,
Thank you very much for your comments concerning our manuscript. We have studied all comments carefully and have made corrections, which we hope will meet with approval.
Revised portions are marked in track changes in the paper.
General comments
The study is well-written and provides valuable information on the seroprevalence and associated risk factors of T. gondii antibodies in pregnant women in Belgrade, Serbia. However, there are several points need clarification from the authors such as;
- In the title, the authors should specify Belgrade in the title, because it is the only tested region in Serbia.
Response: Added.
- In abstract and introduction, the authors should clearly highlight the point of novelty of this study.
Response: We thank the reviewer for this constructive comment. In the Abstract and Introduction, we added the novelty of this study.
-In introduction, the authors should prepare a paragraph for previous reports on seroprevalence and risk factors analysis of T. gondii either in human or animals.
Response: We thank the reviewer for this suggestion. We added the most important data on seroprevalence and explained risk factors.
- In methods, the authors need to add some details on type of sample used and methods of preparation.
Response: Thank you for this important suggestion. We explained that the sample consisted of consecutive women attending the Institute for Blood Transfusion of Serbia for testing before delivery.
- In limitations at lines 214-255, why do authors investigate only pregnant women in Belgrade and what are the efforts done to make this study more comprehensive?
Response: Thank you for this valuable comment. We added a few sentences explaining how we have overcome selection and healthcare access bias in this study. Besides, this was a pilot study. Similar studies focusing on the entire country are planned for the near future.
Minor points
Some typos errors corrections are required as in the following points;
- In title (line 2), the word “Toxoplasma gondii” should be changed to italic, and this should be checked throughout the manuscript.
Response: We apologize for this typographical error. We corrected it.
- Line 23, this “95%CI95 9.1-17.0%” should be corrected and full name should be written because it is the first appearance.
Response: Revised accordingly.
- Line 27, OR should be written in full name because it is the first appearance.
Response: Revised accordingly.
- Line 51, regarding this expression “95% credible interval” did the author mean confidence interval?
Response: Yes, it is the confidence interval. We changed it.
Comments on the Quality of English Language
No serious issues regarding the English edition. Minor revision is required.
Response: Thank you very much for your comments. This manuscript was edited for English language, grammar, punctuation, spelling, and overall style.
Reviewer 2 Report
The paper would be benefited from addressing the following comments and questions:
1/ Species name should be in italics throughout the text.
2/ Line 16 ‘of not if
3/ Line 24 the multivariate logistic regression analysis ...revealed
4/ Line 24… “antibodies is relatively low, there is a need for continuous use and adjustment of evidence-based preventive measures against toxoplasmosis’. This doesn’t give meaning and should be rewritten.
5/ The status and preventive measures for Toxoplasmosis in the country should be reflected in the introduction well and a comparative discussion needs to be done in the discussion part.
6/ How were the participants screened or selected for the serological test? What sampling strategy have you used?
7. Materials and methods should be elaborated further. For example, agglutination assay (HSDA) and the enzyme immunoassay protocols should be discussed and add reference if not your protocol.
8/ What type of informed consent have you received? Is it written?
9/ Conclusion is not complete. From the study finding mention actions to be taken by responsible health authorities to prevent the disease.
needs some work.
Author Response
Dear reviewer,
Thank you very much for your comments concerning our manuscript. We have studied all comments carefully and have made corrections, which we hope will meet with approval.
Revised portions are marked in track changes in the paper.
Comments and Suggestions for Authors
The paper would be benefited from addressing the following comments and questions:
1/ Species name should be in italics throughout the text.
Response: We checked all species names and wrote them in italics.
2/ Line 16 ‘of not if
Response: Corrected
3/ Line 24 the multivariate logistic regression analysis ...revealed
Response: Corrected
4/ Line 24… “antibodies is relatively low, there is a need for continuous use and adjustment of evidence-based preventive measures against toxoplasmosis’. This doesn’t give meaning and should be rewritten.
Response: We thank the reviewer for this suggestion. This sentence is rewritten.
5/ The status and preventive measures for Toxoplasmosis in the country should be reflected in the Introduction well and a comparative discussion needs to be done in the discussion part.
Response: Thank you for this valuable comment. We added one paragraph about preventive measures in the Introduction and compared the situation regarding toxoplasmosis prevention in pregnant women in our and EU countries in the Discussion.
6/ How were the participants screened or selected for the serological test? What sampling strategy have you used?
Response: Thank you for this important suggestion. We explained that the sample consisted of consecutive women attending the Institute for Blood Transfusion of Serbia for testing before delivery.
- Materials and methods should be elaborated further. For example, agglutination assay (HSDA) and the enzyme immunoassay protocols should be discussed and add reference if not your protocol.
Response: We thank the reviewer for this suggestion. We added one paragraph in the Method about HSDA and one reference.
8/ What type of informed consent have you received? Is it written?
Response: Thank you for this important question. We added the manuscript that “The written informed consent form prepared by researchers and approved by the Ethics Committee was filled out by every pregnant woman who agreed to participate in the research.”
9/ Conclusion is not complete. From the study finding mention actions to be taken by responsible health authorities to prevent the disease.
Response: Thank you for this important suggestion. We extended the conclusion.
Round 2
Reviewer 1 Report
The authors responded to the comments and this improved the quality of the manuscript significantly.